# A Fit-for-Purpose Nutrient Profiling Model to Underpin Food and Nutrition Policies in South Africa

**DOI:** 10.3390/nu13082584

**Published:** 2021-07-28

**Authors:** Tamryn Frank, Anne-Marie Thow, Shu Wen Ng, Jessica Ostrowski, Makoma Bopape, Elizabeth C. Swart

**Affiliations:** 1School of Public Health, University of the Western Cape Faculty of Community and Health Sciences, Cape Town 7535, South Africa; 2Menzies Centre for Health Policy, School of Public Health, Charles Perkins Centre, The University of Sydney, Sydney, NSW 2006, Australia; annemarie.thow@sydney.edu.au; 3Department of Nutrition, Gillings School of Global Public Health and the Carolina Population Center, The University of North Carolina, Chapel Hill, NC 27599-7400, USA; shuwen@unc.edu (S.W.N.); jessica.ostrowski@unc.edu (J.O.); 4Department of Human Nutrition and Dietetics, Faculty of Health Sciences, University of Limpopo, Polokwane 0727, South Africa; makoma.bopape@ul.ac.za; 5Department of Dietetics and Nutrition, University of the Western Cape, Cape Town 7535, South Africa; rswart@uwc.ac.za; 6DSI/NRF Centre of Excellence in Food Security, University of the Western Cape, Cape Town 7535, South Africa

**Keywords:** nutrient profiling, South Africa, LMIC, food policy

## Abstract

South Africa (SA) is facing a rising prevalence of obesity and diet-related chronic diseases. The government is seeking to develop effective, evidence-based policy measures to address this. A well-designed, fit-for-purpose nutrient profiling model (NPM) can aid policy development. The aim of this study was to develop a fit-for-purpose NPM in SA. Steps included: (1) determining the purpose and target population; (2) selecting appropriate nutrients and other food components to include; (3) selecting a suitable NPM type, criteria and base; and (4) selecting appropriate numbers and thresholds. As part of the evaluation, the nutritional composition of packaged foods containing nutritional information (*n* = 6747) in the SA food supply chain was analyzed, a literature review was undertaken and various NPMs were evaluated. Our findings indicated that it is most appropriate to adapt an NPM and underpin regulation with a restrictive NPM that limits unhealthy food components. The Chile 2019 NPM was identified as suitable to adapt, and total sugar, saturated fat, sodium and non-sugar sweetener were identified as appropriate to restrict. This NPM has the potential to underpin restrictive policies, such as front-of-package labelling and child-directed marketing regulations in SA. These policies will support the fight against obesity and NCDs in the country.

## 1. Introduction

There is global consensus that better policies are needed to address the obesity pandemic [1,2]. Comprehensive, operationalizable and solidly designed food policies have the capacity to substantially improve diets at a local, national and international level. These benefits are not only for a select few, but also reach disadvantaged, lower socioeconomic groups [3]. Underpinning various country-level food policies with one well-designed nutrient profiling model (NPM) can support good regulatory practice and a consistent regulatory approach through providing a transparent basis on which to distinguish healthier and less healthy foods for policy application [4]. NPMs developed together with industry are more lenient and permit more foods than those grounded with scientific evidence [5], resulting in little progress, if any, in addressing rising rates of malnutrition in all its forms and NCDs. As such, the development of an NPM should be independent of industry involvement.

There are a range of policy measures that can benefit from an NPM as their foundation. These include policies and regulations on front-of-package labelling, marketing to children, school food guidelines, health and nutrition claims on foodstuff, food taxation, food fortification, food procurement in hospitals, prisons and old age homes and the informing of welfare support schemes [6]. Several countries have introduced mandatory food policies that make use of NPMs. For instance, Chile has used the same NPM to effectively restrict foods that have front-of-pack (FOP) warning labels from being sold or eaten on or near school grounds, and to prohibit marketing of foods carrying an FOP label [7].

### 1.1. The Need for Evidence-Based Restrictive Food Policies in South Africa

SA has undergone a transition from infective to non-communicable diseases [8] and South Africans are consuming more and more harmful, ultra-processed foods high in fats, sugar and salt [9]. Given this, the SA government has identified the need to implement policies and practices to prevent and control obesity [10]. Indeed, the current COVID-19 pandemic, with poorer prognosis and higher mortality rates linked to obesity and NCDs [11] has highlighted the strain that obesity places on the healthcare system [12].

The active SA food labelling regulation, R146, was implemented in 2010 [13]. Following this, in 2014, a draft regulation, R429, was published [14] that makes recommendations for an NPM for health and nutrition claims (hereafter referred to as the SA HNC NPM). In June 2016, SA implemented mandatory upper sodium limits in various processed food categories [15]. Regulation 127 of 2011, for trans-fats, prohibits foods that contain more than 2 g of trans-fat per 100 g of oil or fat [16]. These actions suggest that there is recognition of the potential harms associated with certain ingredients or amounts of nutrients in SA and a need to restrict them.

There is a need for strong policies to promote health and prevent non-communicable diseases (NCDs) in low- and middle-income countries (LMICs) [17]. A transparent, evidence-based approach without industry interference should be followed in their development [17,18]. In SA, powerful commercial actors have been shown to influence health policy formulation processes to favor their interests over those of the public’s health and pockets [19,20,21,22]. As such, an independently developed NPM may support the government to develop a strong policy.

### 1.2. Considerations for a Fit-for-Purpose NPM to Underpin Restrictive Food Policy in SA

Different NPMs vary significantly in purpose and complexity, so it is of utmost importance that policymakers consider the purpose and operationalizability of the model they intend to implement, in order to select a model that will achieve the intended purpose [6,23]. Implementation of policy related to NPMs has been slow in LMICs, possibly due to limited resources and a lack of population level nutritional data that are required to inform the development of NPMs within these countries [17,24,25]. Additionally, the LMIC setting faces a number of challenges in the food policy arena, including struggles with multi-sectoral collaboration, implementation of proposed policies and the ability to follow through with long-term commitment to policy goals [26]. 

According to a 2018 systematic review of all NPMs used in government regulation globally (78 models worldwide), only one NPM has been developed for Africa [6]. This is the SA HNC NPM, included in the draft R429 of 2014 for the purpose of regulating health and nutrition claims. It has also been validated for use in child-directed marketing restrictions [27,28]. This model is based on the Food Standards Australia/New Zealand’s (FSANZ) NPM, which in turn was adapted from the UK Ofcom NPM [29], and thus originally designed for high-income countries. Since then, the World Health Organization (WHO) African Region has proposed an NPM [30], although it has yet to be implemented by any country. 

A single fit-for-purpose NPM that can be used to ground various restrictive food policies in SA is an ideal starting point for regulating the processed food environment and will ensure a consistent message for the public and the food industry. Using one NPM in various policies can also reduce administrative burden [4]. If regulations are to be put in place, they need to be easy to implement, require limited resources to enforce and they must not be costly. The objective of this paper is to identify a suitable, context-specific NPM for food policy in SA, using an established step-wise approach.

The LMIC setting of SA offers a diverse and challenging context. It is important that an independent, robust approach following accepted scientific process is used to develop a feasible, context-specific NPM [17,31,32,33]. This paper contributes to existing scientific research on NPMs by investigating the various aspects to consider when developing a fit-for-purpose NPM for restrictive food policies in SA, which has the potential to influence food policy in South Africa, and more broadly, other LMICs in Africa. 

## 2. Materials and Methods

For this paper, the NPM development process was informed by the internationally accepted seven-step approach developed by Scarborough, Rayner and Stockley [32] and the six steps recommended for NPM development or adaptation in the WHO draft guidelines on front-of-pack labelling (FOPL) [33]. Although this is presented as a step-wise process, these steps are practically interrelated and interdependent and decisions made in one step affect the decisions in other steps [34]. Steps will be discussed in combination under these broad headings:Determining the purpose, and target population of the NPM;Selecting appropriate nutrients and other food components to include;Selecting a suitable NPM type, criteria and base;Selecting appropriate numbers and thresholds.

The decisions are informed by a combination of location-specific primary research, as well as lessons learnt through existing literature. Where literature was reviewed, the authors used narrative literature review methodology to identify appropriate, targeted literature. Taking the resource limitations of the country into account, provisions for straightforward classification, implementation and evaluation were considered at each step.

### 2.1. Data

Primary data collection was necessary to evaluate the composition of packaged foods in SA. Nutritional information on packaged foods and beverages (*n* = 6747) with nutrition information panels (NIP) was collected photographically by trained fieldworkers from large supermarkets in SA (Pick ‘n Pay, Woolworths, Checkers, Spar and Shoprite) in 2018. Products that did not contain an NIP were excluded from analysis (Appendix A contain detailed methodological information). These data were used to calculate the mean nutritional content of various nutrients of concern. For non-sugar sweetener (NSS) specifically, where the mean content could not be calculated (as data are not provided), the number of different NSS’s was assessed by identifying NSS in the ingredient list using a standardized list of search terms (Appendix A).

### 2.2. Steps in NPM Development

#### 2.2.1. Step 1: Determining the Purpose and Target Population of the NPM

To determine the purpose and target population, we reviewed relevant population level data [35,36,37,38,39,40,41] and the policy context to identify the key nutritional problems faced by the SA population. We researched existing literature to identify the target population, their disease burden and dietary intake patterns to select the most appropriate NPM. We also reviewed strategic plans and dietary guidelines developed by the SA Department of Health to gain insights into the nutrition problems identified by the government for the population as a whole.

To assess the level of processing in South African packaged foods, collected data were evaluated by making use of the NOVA classification system as described elsewhere [42,43]. 

#### 2.2.2. Step 2: Deciding Which Nutrients and Other Food Components to Include

Making use of literature review, whilst considering the purpose of the NPM and risk profile of the target population, international dietary guidelines and nutrients and food components considered for inclusion in the NPM were reviewed to identify those most appropriate to include. 

As current draft regulations [14] in SA make use of an NPM that includes nutrients of concern to both encourage and limit, in this study we compared differences between an NPM that only considers unhealthy nutrients of concern to restrict, and one that includes both healthy and unhealthy components by applying their criteria to the SA packaged food supply (*n* = 6747). The Chile Warning Octagon (CWO) 2019 was selected as it assesses nutrients of concern linked only to poor health outcomes (sugar, sodium, saturated fat and energy) that need to be restricted through food policy. A number of studies indicate its success in Chile [44,45,46,47,48], and other countries have used Chile’s NPM [49,50]. The SA HNC NPM assessed health and nutrition claims in draft SA regulation [14] and was adopted from Food Standards Australia and New Zealand’s (FSANZ) NPM (which relied on the UK Ofcom NPM in its development [29]). To determine the difference between the SA HNC NPM baseline score (which is the score solely for ‘nutrients to limit’—sugar, sodium, saturated fat and energy) and the final SA HNC NPM score (which includes points for fiber, protein, fruits, vegetables, nuts and legumes as ‘nutrients to encourage’), we evaluated the baseline and final SA HNC NPM separately, alongside the CWO 2019 NPM. 

Algorithms were generated in STATA (version 15, StataCorp, College Station, TX, USA) to evaluate the packaged food supply against these NPMs. We classified foods and beverages as either compliant or non-compliant, depending on the nutritional criteria of the NPM. For the SA HNC NPM, a product was considered compliant when it met the criterion of being allowed to carry a health claim (for beverages a score of less than 1, for processed cheese and fats a score of less than 28 and for other foods a score of less than 4). Similarly, for the SA baseline HNC NPM, a product was considered compliant when it met the criterion of being allowed to carry a health claim—but for this model, points were not awarded for ‘nutrients to encourage’. For the CWO 2019 NPM, a product was considered compliant when it was excluded from carrying a warning label according to the 2019 criteria (it did not exceed any of the model’s limits for energy, sugar, sodium or saturated fat).

A fruits, vegetables, nuts and legumes (FVNL) score needed to be calculated for the SA HNC NPM. As the percentage of fruits and vegetables is not routinely included on the NIP in SA, we manually estimated scores. A similar methodology, as described by Bernstein et al. [51], was followed for the calculations. An FVNL score of 0 was assigned to sub-categories without FVNLs (e.g., fats and oils). For groups which may contain FVNLs, products were individually reviewed within the context of its group. We considered the order of ingredients, the form of the FVNL ingredients (concentrated or non-concentrated), number of FVNL ingredients compared to the number of non-FVNL ingredients and type of product in order to determine the score. A dietitian performed all classifications.

To consider nutrients of concern within packaged foods in SA, we divided the packaged food supply into sub-categories and the mean content of nutrients of concern was calculated. Most of this information was available from the NIP, however, free sugar content of products was estimated according to the method proposed by the Pan American Health Organization (PAHO) [52] when not available on the NIP. To evaluate the suitability of the inclusion of energy in the NPM products excessive in energy only (and not sugar, saturated fat or sodium), the CWO 2019 NPM was utilized. 

#### 2.2.3. Step 3: Selection of a Suitable NPM Type, Criteria and Base

To inform the selection of the NPM type, we used the approach described by Rayner, Scarborough and Stockley [31,32] to decide on the base unit of measure, whether to make use of a categorical or continuous model, as well as whether to select across-the-board thresholds or category-specific criteria. 

The criteria underpinning an NPM can be applied across all foods, (‘across-the-board’) or alternatively, it can differentiate between food categories, allowing either comparison across different food categories or comparison within food categories (‘category-specific’) [53]. Models can be either threshold-based or continuous [32]. Threshold-based NPMs identify predetermined cut-points that will place food into different categories in a binary manner. Different cut-points can be used for different nutrients in one food (e.g., a different cut-point each for sugar, sodium and saturated fat). With a continuous model, foods are classified on a continuous scale, ranking nutritionally better, or worse, than another product. Points are awarded based on different criteria for various nutrients and a summative healthfulness score is calculated from this, depending on the NPM’s criteria [31]. 

To inform the decision, we conducted a review of the literature, examining the pros and cons of various approaches, whilst taking the LMIC context of SA into consideration.

#### 2.2.4. Step 4: Choosing the Thresholds to Use

In order to decide whether to use the same cut-points as an existing NPM, or to adapt them for SA, we followed the same methodological approach as that described in the Chilean paper [54] for the development of cut-points. For the SA list of foods, two dietitians analyzed the SA Medical Research Council Food Composition Table (FCT) [55] independently of each other. Any discrepancies in classification were resolved in a group discussion with a third dietitian. All three dietitians were in agreement about the final list of foods included. As the SA FCT is not as comprehensive as the USDA FCT, the final list of foods was shorter, at 183 foods.

## 3. Results 

### 3.1. The Purpose and Target Population of the NPM

South Africans are consuming more and more harmful, ultra-processed foods [9,56] that are linked to NCDs [57]. This is clear from the analysis of the packaged food supply in SA (Figure 1 and Appendix A). In 2018, 83% of packaged products evaluated in SA supermarkets (89% of foods and 61% of beverages) were either processed or ultra-processed, according to the NOVA classification [42,43].

The NCD mortality rate has steadily increased and NCDs are the main cause of death in SA [58]. The 2016 SA Demographic and Health Survey (SADHS) reported that overweight, obesity and hypertension prevalence have been on the rise since 1998, with 31% of men and 68% of women overweight or obese [35]. A recent report by the World Obesity Federation anticipates that SA is likely to have the 10^th^ highest level of childhood obesity in the world by 2030, with approximately 28% of children aged 5 to 19 obese in 2030 [36]. 

Additionally, a review of all adult dietary studies done in SA between 2000 and 2015 found that there is still a high prevalence of micronutrient deficiencies. High food prices and limited availability of nutrient-dense foods in townships and poorer urban centers are thought to be contributing factors [37]. A predominantly carbohydrate-based diet with low nutrient density is common in SA [38,39,40,41]. 

The nutrition policy priorities in SA are clearly articulated in the National Department of Health’s Strategy for the Prevention and Control of Obesity in SA 2015–2020: ‘create an enabling environment that supports the availability and accessibility of healthy food choices in various settings’. It highlights the need for the development of norms and standards on sugar and fat content of ultra-processed foods and also notes the importance of FOPL and consideration of ethical marketing of food to children [10]. Currently, draft regulation [14] in SA includes an NPM to assess health and nutrition claims, but no NPM has been developed for SA to use in restrictive food policies, such as FOPL and marketing to children.

Although those at risk for developing NCDs and children (with rapidly increasing obesity prevalence) will benefit most from the potential restrictive food policies that the NPM can underpin, these food policies are broadly applicable across different population groups and age groups. Due to heterogeneity in nutritional requirements across different life stages, it is recommended that adult dietary guidelines be used to guide NPM criteria [33]. The only age group excluded from NPMs is children below the age of six months, where exclusive breastfeeding is recommended and protected by existing regulation in SA [59].

#### Recommendation

The purpose of the NPM should be to identify unhealthy, processed packaged foods that result in poor health outcomes. The appropriate target population for the NPM is all adults and children above the age of six months.

### 3.2. Nutrients and Other Food Components to Include

Literature review indicated that countries implementing regulations (with mandatory NPM applications) have predominately focused on negative nutrients of concern linked to poor health outcomes, which they restrict through policies such as FOPL, marketing restrictions and regulations in the school environment [18]. This is because generally, using an NPM that includes both ‘nutrients to limit’ (such as salt, sugar and saturated fat which are harmful to health) and ‘nutrients to encourage’ (such as fiber, fruit and vegetables, nuts and legumes which are beneficial to health), in a restrictive regulatory environment is limited and becomes complicated to implement. To date, only Israel has added another, positive FOPL in the form of a green logo for healthy, minimally processed foods [60] that do not contradict their warning labels. 

From the results in Figure 2 and Table 1, it is apparent that the additional points awarded for ‘nutrients to encourage’ in the final SA HNC NPM give rise to a more lenient score than the initial SA HNC NPM baseline score, which only includes ‘nutrients to limit’, resulting in more compliant products in all categories, except for sodas, where it does not change. These results highlight the leniency that ‘nutrients to encourage’ introduces into the final score of an NPM. Of the models included in this study, the most lenient model was the SA HNC NPM, with 47% of foodstuffs compliant, and the SA HCN Baseline model was the strictest, with 19% compliance. 

A review of the literature indicates that trans-fat, saturated fat, added or free sugar and sodium are harmful to health, and contribute to the obesity and NCD epidemic [61,62,63,64,65,66]. According to a review of NPMs used in government-led nutrition policies, all NPMs (*n* = 78) included ‘nutrients to limit’. The most common nutrients of concern identified in these NPMs were sodium, saturated fat and total sugar [6]. Table 2 provides the mean nutrient content of unhealthy nutrients of concern in packaged foods in SA.

#### 3.2.1. Sugar

International guidelines recommend the restriction of free or added sugar (rather than total sugar) as this is the sugar that is harmful to health [67]. Please refer to Appendix A for the definitions of sugar used in this paper. Intrinsic sugar is not believed to be as harmful as free sugar [67]. These differences can be seen in Table 2 for the dairy and fruit categories, where the total and free sugar content differ significantly due to the high intrinsic sugar levels. 

WHO dietary guidelines recommend that a maximum of 10% of daily energy should be derived from total sugar and 5% from free sugar [67]. Based on a reference energy intake of 8400 kJ (or 2000 kCal) per day, this equates to 1 g of total sugar per 168 kJ and 1 g of free sugar per 336 kJ consumed in a day. As the average energy content per 100 g of foods in this sample is 1072.8 kJ (see Table 2 above), in order to align with WHO dietary guidelines, total sugar should ideally be below 6.4 g/100 g and free sugar should be below 3.2 g/100 g. Similarly, for beverages (with an average energy content of 160.7 kJ/100 mL), total sugar should be below 1 g/100 mL and free sugar below 0.5 g/100 mL. However, the sugar content of foods in this sample far exceeds the sugar recommendations in the WHO dietary guidelines, as 21.2% of energy (13.4 g/100 g) is attributed to total sugar and 18.4% (11.6 g/100 g) to free sugar. Although beverages contain less sugar per 100 mL than foods, sugar contributes most of the energy in beverages, at 76.2% (7.2 g/100 mL) and 54.0% (5.1 g/100 mL), respectively, for total and free sugar.

As the chemical structure of free (or added) sugar is the same as total sugar they cannot be differentiated objectively through laboratory tests [68]. One could require manufacturers to report the added sugar on the NIP (as in the USA), although this is reliant on the manufacturer being trustworthy about the recipe composition. Alternatively, the approximate amount of free or added sugars in foods can be calculated [52,69]. However, this approach requires assumptions to be made, is time-consuming and open to misinterpretation. In the resource-limited LMIC setting, this is not a suitable method to use. The total and added sugar challenge is not new. When the UK Office of Communications (Ofcom) NPM was being developed, added sugar was proposed, but due to the technical difficulties involved in analyzing added sugars, total sugar was selected instead [70]. Numerous other countries, including Chile [7], Israel [50] and Peru [49], have opted to use total sugar in their regulations.

In order to work around this, Chile recommends applying the total sugar cut-point of the NPM only to those foods that have added sugar, salt or saturated fat [54]. This is to prevent foods such as fresh fruit from receiving a classification of ‘high in sugar’ (as one would not want to accidentally restrict a healthy product, such as fresh fruit, through the application of an NPM). However, this allows 100% fruit juice to ‘pass’ the CWO 2019 NPM criteria as it is considered free, but not added, sugar (see Table 3 below in which 99 percent of 100% fruit juices are compliant with the CWO 2019 sugar criteria). 

This is of some concern, as long-term overconsumption of fructose (the sugar found in fruit juice) may result in cardiovascular and metabolic diseases [71] as well as increased all-cause mortality risk [72]. It has been argued that efforts to reduce sugar consumption need to be extended to 100% fruit juice [72]. In 2019, the Indian Academy of Pediatrics recommended fruit juice should not be given to infants and young children below the age of two years, and that very limited amounts should be given to older children [73]. 

Given the difficulties of measuring added or free sugar, it is recommended that SA make use of total sugar in the NPM. However, unlike Chile, where added sugar is used as a qualifying criterion, it is recommended that the SA NPM use free sugar—including any form of fruit juice concentrate (e.g., pulp)—rather than added sugar in the qualifying criteria.

#### 3.2.2. Fat

Consideration should be given to total fat, saturated fat and trans-fat.

Total Fat: The WHO recommends an intake of between 15 and 30% of total energy from fat [74]. Recently, the US dietary guidelines removed total fat as a nutrient of concern to focus instead on unhealthy saturated fats [75].

Total fat has not been identified as appropriate to include in this NPM because fat is not harmful to health per se. Indeed, certain components, such as mono- and polyunsaturated fatty acids are beneficial to health and provide protection against certain NCDs, such as cardiovascular disease [76,77,78,79]. 

Saturated fat: The WHO recommends that less than 10% of the total daily energy intake should be from saturated fats [80]. The Codex Alimentarius non-communicable disease guidelines for saturated fat (NRV-NCD) recommend that saturated fatty acid intake should not exceed 20 g/day, based on a reference energy intake of 8400 kJ (or 2000 kCal) [81].

The Heart and Stroke Foundation SA, as well as international organizations such as the American Heart Association and Heart UK, recommend limiting saturated fat intake due to the risk of elevated cholesterol levels and the increased risk of heart disease. As saturated fat is known to be harmful to health [76,77,78], it is recommended that it should be included in the NPM. 

Trans-fat: In line with WHO recommendations, SA implemented a regulation on trans-fat, R127, in 2011 [16]. The WHO guideline recommends that less than 1% of the total energy intake be derived from trans-fat [80]. The R127 effectively deals with trans-fats and prohibits foods sold in SA to have more than 2 g of trans-fat per 100 g of fats and oils. The analysis of the mean trans-fat content of packaged foods in SA (Table 2) indicates that the mean trans-fat level is 0.09 g, which is below 1 g per 100 g. Thus, it is unnecessary for the NPM in SA to include trans-fat as the country has already effectively dealt with this harmful nutrient through regulation R127 of 2011. 

#### 3.2.3. Sodium

Salt should be restricted to less than 5 g per day, and sodium to less than 2 g per day according to WHO Guidelines [82]. Codex Alimentarius recommends that sodium intake should not exceed 2000 mg per day, based on a reference energy intake of 8400 kJ (or 2000 kCal) to prevent NCDs. This translates to 1 mg per 4.18 kJ [81]. SA introduced sodium regulations (R214/2013 [15]) in two phases, from 2016 to 2019. The intended purpose of this regulation is to reduce sodium levels in foods with the aim to reduce hypertension, and is category specific and does not target all foods [83]. 

As the mean energy content in SA packaged foods (Table 2) is 1072.8 kJ/100 g, one would expect the sodium content to be below 258 mg/100 g to align with recommended dietary intake [81,82]. However, the mean sodium content is high, at 411 mg/100 g. There are four food categories: mixed dishes; protein; snack foods; and soups and sauces, which remain particularly high in sodium. Unlike the trans-fat regulations, which adequately address trans-fat by virtually removing it from the SA market, sodium needs to be included in the NPM as certain products have excessive quantities of sodium.

#### 3.2.4. Energy

Although Chile [7] and the WHO African Region [30] have opted to include energy as a criterion in their NPMs, it is postulated that packaged and processed foods high in energy are also high in sugar, saturated fat and/or sodium. Further, there is an expectation that by including criteria for sugar, saturated fat and sodium, most foods high in energy will be addressed. 

This was found to be true when applying the thresholds for the CWO 2019 NPM to the SA packaged food supply (Table 3). Very few products were excessive only in energy. Of the 4344 products regulated, only 100 products (or 2.3%) were above the cut-off for energy, but no other nutrient. The other 97.7% of products were regulated for sugar, sodium and/or saturated fat. This is in line with Camacho and Ruppel [84] who argue that by focusing on the calorie balance (total energy) in policies, one gives the food industry a convenient exit strategy so that they can avoid engaging with the obesity crisis. Diet composition, particularly in the case of processed foods, is potentially more harmful to health than the overall calorie balance. In countries facing high levels of stunting, wasting and micronutrient deficiencies, NPMs that focus on energy may be problematic [85].

#### 3.2.5. Non-Sugar Sweetener

For this paper, following the NPM for the WHO Africa Region [30], the term non-sugar sweeteners (NSS) will be used (definition in Appendix A). The use of NSS, such as artificial sweeteners and polyols, in the food supply is becoming more commonplace and has become central to sugar substitution [86]. They are consumed not only through foods, but also in medicines, food supplements and other products such as toothpaste [87]. NSSs are among the most widely used food additives globally [88]. This is partly because consumers are interested in reducing sugar intake [89] and also because the introduction of food policies such as a sugary beverage tax or front-of-pack labelling has incentivized the food industry to reformulate and replace sugar with NSS [90,91] instead of reformulating into products that are less sweet. Table 2 indicates that 55.6% of sodas (and 29.6% of all packaged beverages) and 12.8% of confectionery (and 6.6% of all packaged foods) in SA already contained NSS in 2018 (before the implementation of South Africa’s sugar-sweetened beverage tax known as the Health Promotion Levy).

Because of the increased use of NSS, it is important that a thorough evaluation be conducted of its risks and benefits before advocating for, or discouraging, its use [92]. Although numerous studies and systematic reviews of these studies have been conducted on the topic of safety in the use of NSS, there is no consensus among researchers. One of the challenges is that many different NSSs exist and new NSSs are constantly becoming available [89]. 

The sweetness level of different NSSs also differs [93,94]. They do not all have the same physiological effect [95], and quantities of NSS intake are largely unknown. Worldwide, only two countries (Chile [7] and very recently Saudi Arabia [96]) include quantities of each type of NSS on the NIP. This makes it nearly impossible to accurately investigate consumption volumes across the world as the data are simply not available. 

In Chile, food companies are reformulating products to replace sugar (which is regulated) [91] with NSS (which is not regulated). More than half (55.5%) of all packaged products in Chile now contain at least one NSS, making it difficult to select NSS-free options [97]. In SA, since the introduction of the HPL, there has been growing evidence of product reformulation [98], and although NSSs have yet to be investigated, many brands have reformulated sugar down from above 10 g/100 mL to less than 5 g/100 mL [99]. It is likely that much of this sugar has been replaced with NSS.

It is impossible to set a cut-point for NSS, unlike for other nutrients of concern such as sodium, saturated fat or sugar which have evidence-based cut-points, as there is currently inadequate evidence to identify an NSS cut-point. However, there is growing concern around children’s exposure to NSS and its effects on their sweetness preferences later in life [100,101] and gut health [102]. The impact of prolonged use remains unclear [102], and recently, NPMs targeting children have recommended that children’s exposure to sweeteners be restricted [30,103].

As food policies should be proactive in protecting the health of the population, the evidence currently available suggests that it is wise to regulate the use of NSS or at the very least, require clear information about its presence and amounts in food products to monitor its presence better. After all, the purpose of food policies is to encourage a shift towards the consumption of more whole, unprocessed foods rather than alternative, ultra-processed foods. 

#### 3.2.6. Recommendation

An easy to operationalize NPM aimed at reducing the demand for processed foods linked with NCDs and obesity should include ‘nutrients to limit’, and exclude ‘nutrients to encourage’. Saturated fat, sodium, non-sugar sweetener and total sugar (with free sugar, not added sugar, used as the qualifying criteria when assessing the inclusion criteria of the NPM) have been identified as appropriate to include in the NPM.

### 3.3. Selecting the NPM Type, Criteria and Base

Base unit of measure: WHO and Codex Alimentarius dietary guidelines provide guidance with regard to nutrients of concern in reference to their contribution to the percentage of total energy [67,80] or as a nutrient reference value that should not be exceeded per day [81,82]. However, these refer to the total daily intake per person, and it is not easy to practically implement because different people have different energy requirements, and packaged foods represent only a portion of overall daily intake. The draft WHO FOPL guidelines recommend NPMs be developed using a per 100 g approach [53]. The portion size or per serving approach results in several challenges as different age groups should have different portion sizes, and consumption patterns differ among individuals [70]. Portions are easier for the food industry to manipulate, and often represent portion sizes that are ‘healthier’ but not realistic in relation to the package size or the amount that people eat. Consistent with Codex Alimentarius guidelines [81], nutritional information in SA is displayed in a per 100 g/100 mL format for foods and liquids. Considering the pros and cons of various options available (Table 4), and given the current 100 g/100 mL format used in SA, continuing in the same manner would be practical.

Across the board vs. food category specific: An across-the-board approach establishes consistent criteria which limit the risk of misinterpretation or incorrect classification [6]. It is not resource-intensive and is straightforward to implement. However, as all foodstuffs are treated in the same manner, regardless of their inherent nutrient composition, it could suppress reformulation within a category if changes are needed for most of the foods in that category. A category-specific approach, such as is used for marketing restrictions by the WHO Africa Region NPM [30], allows for criteria that are specific to the nutritional composition of different types (or categories) of food; and the criteria can be informed by the nutritional content of existing foods in the category [6]. However, the numerous categories with different thresholds make it difficult for regulators to implement and it potentially allows leeway for the food industry to manipulate within category thresholds. 

It is important to consider the context, and to weigh up robustness with the ability to apply it appropriately when selecting the most appropriate NPM [6]. To date, all countries that have adopted a mandatory warning label model have focused on only two categories (food and non-alcoholic beverages). This includes Chile [104], Israel [50], Peru [49] Uruguay [105], Mexico [106] and most recently, Brazil [107]. To ensure a simple, easy-to-implement approach, a category-specific approach is not appropriate for the SA setting as it is more resource-intensive (both for implementation and evaluation). An across-the-board approach is recommended for the NPM.

Threshold vs. continuous: A threshold-based approach has been successfully implemented in the mandatory, restrictive food policies of a number of countries, including Israel [50], Chile [104] and Peru [49]. It is administratively simple, as no calculation or comparisons need to be made before classifying a food (it either meets the cut-point or it does not). Previously, when the generally encouraged model was one that included both nutrients to encourage and limit, a continuous or scoring system was well justified. The rationale was that foods were composed of many nutrients and a single cut-off would result in the loss of valuable information [70]. However, this argument only holds true when both nutrients to encourage and restrict are considered or when the NPM is being used to underpin a positive logo that focuses only on whole, minimally processed foods, as in Israel [60]. 

Continuous models require a number of different calculations to be performed and can be human resource-heavy. Ultimately, when used together with an FOPL or health claim system, a threshold is still used to determine whether a product can carry a claim or not, or whether it is red, yellow or green. In this sense, a scoring approach is always used in conjunction with a threshold.

#### Recommendation

The SA NPM should use a straightforward approach and an across-the-board, threshold-based approach, that is applied to all packaged foodstuffs and uses a ‘per 100 g’ base for solids and the ‘per 100 mL’ base for liquids. 

### 3.4. Thresholds to Use

The most appropriate and relevant nutrient cut-points are selected based on the recommendations made in steps one through three. 

Where possible, it is recommended to adapt existing NPMs to make them context-specific, rather than inventing them from scratch due to the immense time and resources required to develop an NPM [33]. Based on the above considerations, the CWO 2019 NPM [7] appears to be the most suitable NPM to adapt. For the purpose of this paper, other NPMs considered include Mexico [106], Peru [49], Israel [50], PAHO [52], WHO Africa Region [30] and the FSANZ (SA HNC) [14] models (Appendix A). The PAHO NPM considers the percentage-of-energy approach, rather than a per 100 mL or per 100 g approach, and it is therefore not appropriate. The SA HNC NPM considers both nutrients to encourage and limit, which is difficult to implement and does not meet the purpose as discussed earlier. The WHO African Region NPM uses a category-specific approach rather than an across-the-board approach, which is why it has not been selected. Compared to other NPMs considered, the CWO 2019 NPM most appropriately meets requirements based on recommended components for the SA FOPL NPM. It is a mandatory, threshold-based, across-the-board model using a ‘per 100 g’ or ‘per 100 mL; approach and it focuses on negative ‘nutrients to limit’. A number of other countries have already adopted this NPM into their regulation, including Peru [49] and Israel [50]. Furthermore, the Chilean NPM has shown some promising results in the food policies it is underpinning in Chile [44,45,46,47,48].

The cut-points developed for the CWO 2019 NPM were based on nutrient composition analysis of 358 whole, unprocessed foods, using the USDA nutrient database [54]. No similarly comprehensive database is available for SA. The SA FCT includes 183 whole, unprocessed foods (110 foods with nutrients analyzed in SA, 65 based on the USDA FCT and eight on FCTs from other countries), resulting in an SA-specific sample size 69.3% smaller than that used for Chile. Given this, it was deemed appropriate to adopt the Chilean cut-points as-is for sodium, saturated fat and total sugar. This approach was also used by Israel and Peru [49,50].

#### Recommendation

This analysis indicates that the following cut-points (Table 5) be used for the proposed NPM, that have been adapted from the Chilean approach.

It is recommended that this NPM, if used, should be applied to all packaged foods and non-alcoholic beverages in SA containing any of the following:Free sugar;Added sodium;Added saturated fat;Non-sugar sweetener.

## 4. Discussion

Given the proposed NPM’s ability to identify unhealthy products, it is appropriate for use to underpin restrictive food policy in South Africa. This could include FOPL regulations (which indicate packaged products high in unhealthy nutrients of concern), marketing restrictions, taxation policies and policies that restrict unhealthy foods in schools, hospitals and other government institutions. Elsewhere in the world these restrictive policies have been successful initiatives towards promoting a healthier food environment by supporting a move away from unhealthier food choices. [108]. 

The use of an evidence-based NPM built on a scientific basis that supports non-discriminatory policy measures is necessary in the international trade context [109,110], where limitations imposed by international trade and investment agreements have been found to impede public health policies [111]. A scientific basis of measure in policy development ensures trade and investment agreements are respected and do not place undue limitations on public health priorities [109,112]. Without adequately researched, evidence-based regulations, governments run the risk of being forced to retract regulations due to trade and investment agreements, as was seen with the turkey tail ban in Samoa [113]. 

Taking the resource limitations of SA into account, an important consideration in the development of this NPM was straightforward classification, implementation and evaluation for any NPM that is used in national regulations. For example, in our evaluation of the SA HNC NPM, the FVNL score had to be calculated manually for each product as it is not routinely reported on NIPs in South Africa. This was a time-consuming task, performed by nutrition experts. From a monitoring and evaluation standpoint, it is not a feasible assessment approach (due to the time and skills required) and it is not advisable to include an FVNL score in an NPM used in regulation in South Africa without mandating an FVNL nutritional declaration on the product packaging. This principle applies to all food components included in an NPM. It would be prudent, if including NSS in the NPM, to follow Chile [7] and Saudi Arabia [96], and strengthen current sweetener regulations in South Africa [114] by requiring mandatory reporting of NSS quantities on nutritional packaging. One of the newer approaches to NSS policy can be found in Mexico [106]. Their new law on front-of-package profiling includes an information box that states: ‘Contains non-sugar sweeteners, not recommended for children’. This is placed in a large black box with white characters. They originally intended having a warning label for NSSs but, based on industry objections and World Trade Organization concerns, they shifted to an information box which has passed legal scrutiny. 

As with most NPMs, the proposed NPM cut-points are applicable to foods and beverages in an as-consumed form. Should an NPM be used in regulations in SA, food labels should include the nutritional composition per 100 g ‘as-consumed’ alongside the ‘as-packaged’ composition if reconstitution through home preparation is required (e.g., concentrated fruit drinks). Manual calculations are time-intensive and may result in errors. Future regulations should stipulate that, should the ‘as-consumed’ information be missing, the NPM criteria will be applied to the ‘as-packaged’ information to encourage the ‘as-consumed’ information to be included. In addition, should nutritional information be unavailable on the product packaging (or missing for certain nutrients of concern), then by default the product should be assessed as ‘excessive’ in the nutrient of concern for which there is no information.

Distinguishing between healthy and unhealthy foods to regulate through policy is challenging for policy makers as the food industry contests definitions and argues that they are vilifying foods by differentiating them. Because of this, it is important that the purpose of the NPM is clearly understood. Recently, the argument has been made that NPMs in countries with high levels of malnutrition and stunting should include ‘nutrients to encourage’ [85]; however, this was not identified as suitable for the purpose of our proposed NPM. Positive components do not neutralize the negative health consequences of consuming the unhealthy components in the same product. The SA HNC NPM included in this analysis was developed to allow health and nutrition claims, which focus on ‘nutrients to encourage’. However, research by authors involved in the development of the SA HNC NPM concluded that the NPM was more lenient than other NPMs when marketing restrictions were applied to foods high in fat, sugar and salt [27]. This supports concerns that the addition of (healthy) ‘nutrients to encourage’ can confuse the matter when trying to identify unhealthy foods [23]. The approach of classifying foods as ‘healthy’ or ‘healthier’ has allowed the industry to add nutrients or additives (e.g., isolated or synthetic non-digestible carbohydrates that count towards fiber) to otherwise unhealthy products [115]. ‘Healthier does not necessarily mean healthy per se, and the notions of ‘better than’ may mislead consumers away from what is best’ [70]. An NPM with the purpose of identifying unhealthy products to restrict should thus only include ‘nutrients to limit’.

However, there is still space to consider ‘nutrients to encourage’. Work has already been done in SA in the development of a positive FOP logo [116] and an approach similar to Israel could be considered, where additional criteria are used to identify healthy, minimally processed foods that may carry a positive FOP logo [60]. Although we have not identified the current SA HNC NPM proposed in draft R429/2014 as appropriate for restrictive food policy, it could still be useful for its intended purpose—to regulate health and nutrition claims on packaged foods. It is possible that the two NPMs could work in tandem; permitting foods that are not identified as unhealthy via the SA FOPL NPM to be assessed for eligibility of health and/or nutrition claims via the SA HNC NPM. If this approach is considered, we recommend that it be designed carefully to complement the restrictive NPM, only permitting positive messaging on products not carrying an FOP warning label. Having both a warning label and positive message could provide a mixed message and confuse consumers. 

### Limitations and Strengths of the Study

No NPM is without limitations. Consumption frequency, as part of normal daily dietary intake was not considered, and instead we focused on the nutrient profile of the foods themselves, rather than their role in the diet. Nationally representative dietary intake data in SA are sparse, as nutrition surveys are not regularly done, nor are they representative of the population [37]. This makes assessing dietary intake in the country challenging. 

Similarly, certain assumptions, which may not have always been accurate, had to be made as free sugar and FVNL values were not available. Data were collected in the Western Cape Province of SA, at big retail outlets, so it is possible that products that only appear in certain locations or shops were excluded. 

A large number of products were excluded from analysis as the nutritional content of packaged foods could not be assessed for compliance if they did not have an NIP, which is not a legal requirement in SA. To overcome this, the SA government could regulate NIPs on all packaged foods as mandatory. The information this panel provides can be used to assess compliance with various food regulations. In accordance with Codex guidelines [81], the panel should be clear and easy to understand and presented in a standardized manner.

After an NPM has been proposed, it is important that it be validated and tested for appropriateness by applying it to the local food supply chain [117,118]. This was outside the scope of this paper, although it is an essential step before an NPM can be accepted as appropriate for a certain setting. This has been mitigated to some extent by consulting with dietitians on the nutrient thresholds in South Africa, following the approach used in Chile. The validation study has been submitted elsewhere (currently under consideration for publication). 

This study makes use of a recently collected dataset, which, to our knowledge, is the most comprehensive dataset of nutritional information on packaged foods in South Africa. A systematic process was followed to assess various NPM options using available data and literature, as well as existing regulations. Cultural dimensions are often overlooked in policy making processes. Adapting an NPM for a specific context, as has been done for South Africa throughout this NPM development process, is critical to ensure context-specific solutions.

## 5. Conclusions

This article proposes a fit-for-purpose NPM that is suitable to use in restrictive food policies in SA. It is adapted from the Chilean NPM and includes criteria for sodium, total sugar, saturated fat and non-sugar sweetener. It has the potential to be the foundation for FOPL and child-directed marketing regulations, broader taxation policies and to act as a guide for products to prohibit them from being sold or served in schools, hospitals or other government facilities. Although implementing these policies will not resolve the obesity and NCD crisis in the country, they will be an additional step in the fight. It has the potential to inspire other LMIC in Africa and can be scaled up for use elsewhere.

## Figures and Tables

**Figure 1 nutrients-13-02584-f001:**
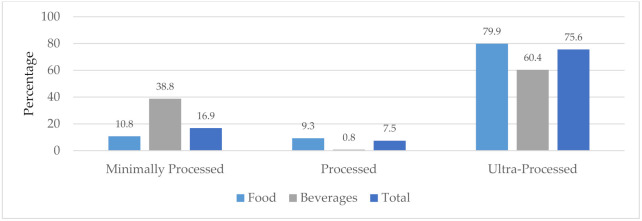
Packaged foods and beverages in the SA marketplace (2018) classified as minimally processed, processed and ultra-processed, according to the NOVA classification system. Note: the NOVA category, ‘culinary ingredients’ has been omitted from analysis.

**Figure 2 nutrients-13-02584-f002:**
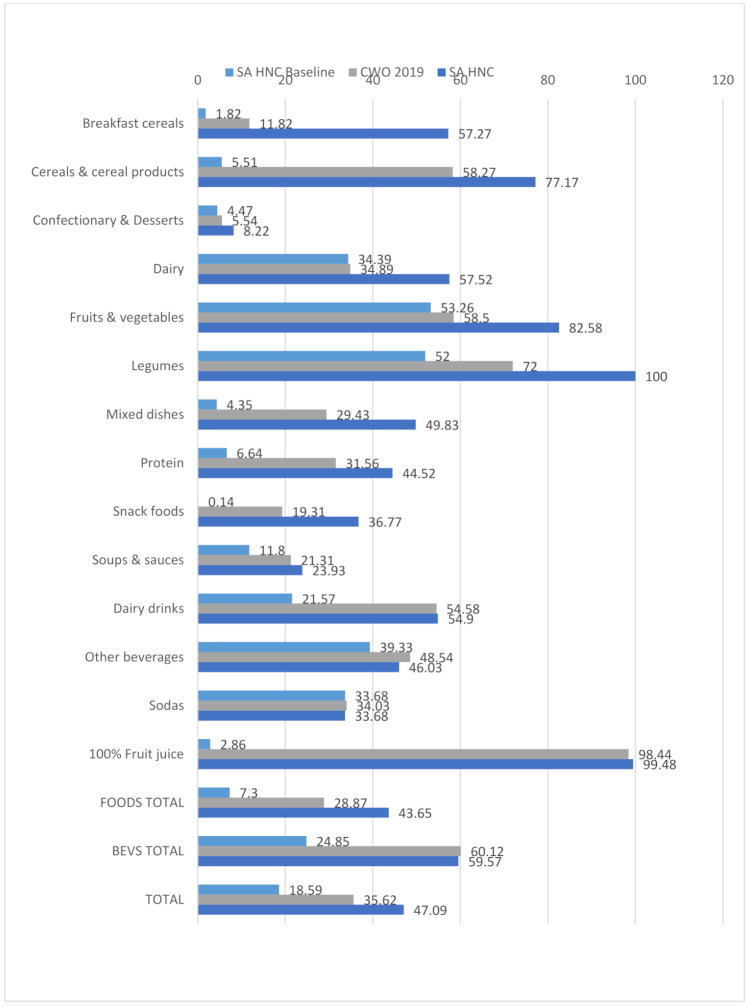
Proportion of SA packaged food and beverage products compliant with the SA HNC, SA HNC baseline and CWO 2019 NPM criteria by groups.

**Table 1 nutrients-13-02584-t001:** Qualitative table reflecting how the addition of ‘nutrients to encourage’ affects NPM leniency, in terms of the proportion of products compliant (i.e., that fall beneath the thresholds and are not deemed ‘unhealthy’).

	Highlighted Food Categories	SA HNC Baseline (Only ‘Nutrients to Limit’	CWO 2019 (Only ‘Nutrients to Limit’)	SA HNC (Nutrients to Limit and Encourage)
Categories where the addition of ‘nutrients to encourage’ may potentially be beneficial	Dairy Drinks	21.57% compliant with criteria	54.6% compliant with criteria	54.9% compliant with criteria
	In this category one would want to mitigate the effects of lactose, a carbohydrate naturally present in milk, which contributes to the total sugar score. Here the addition of ‘nutrients to encourage’ potentially assists; although given the differences in algorithms, the CWO criteria (which only includes ‘nutrients to limit’) has a similar compliance level.			
Categories where the addition of ‘nutrients to encourage’ allows for a more lenient score	Breakfast cereals	1.8% compliant with criteria	11.8% compliant with criteria	57.3% compliant with criteria
	Dairy (food)	34.39%	34.89%	57.52%
	Fruits and vegetables	53.26	58.5%	82.58%
	Legumes	52%	72%	100%
	These categories highlight where the more lenient score of the SA HNC NPM causes contestation due to the addition of protein, fiber, FVNL (fruit, vegetables, nuts and legumes) points. In all of these categories the SA HNC NPM scored at least 23% higher than for the CWO 2019. Although these foods may include healthy components, the impact of undesirable ingredients cannot be negated by ‘nutrients to encourage’. Most of the fruits, vegetables and legumes that are restricted by the CWO 2019 contain high levels of sugar or sodium. The significant change in compliance due to the addition of ‘nutrients to encourage’ is seen in the difference in scores between the baseline and final SA HNC NPM.			
Categories that are strict regardless of nutrients to encourage or limit	Confectionery & Dessert	4.47%	5.54%	8.22%
	Soda	33.68%	34.03%	33.68%
	These categories contain a large number of unhealthy products that are linked to poor health outcomes. Even when ‘nutrients to encourage’ are added, they score poorly across different NPMs. The addition of ‘nutrients to encourage’ does not result in a more lenient score.			

**Table 2 nutrients-13-02584-t002:** Mean content of nutrients of concern in packaged foods in SA per 100 g (2018).

	Energy (kJ)	Total Sugar (g)	Free Sugar (g)	Total Fat (g)	Saturated Fat (g)	Trans Fat (g)	Sodium (mg)	Contains NSS *n* (%)
Breakfast cereals *n* = 110	1588.2	17.2	16.3	8.9	2.9	0.03	210.6	0 (0)
Cereals & cereal products *n* = 254	989.9	3.1	3.2	6.1	2.4	0.11	338.5	8 (3.2)
Confectionary & dessert *n* = 1119	1559.8	38.4	35.1	14.0	7.7	0.12	142.5	143 (12.8)
Dairy *n* = 791	766.5	6.4	3.5	12.6	8.8	0.33	322.1	70 (8.9)
Fruits & vegetables *n* = 196	677.4	29.8	15.2	2.1	1.2	0.02	41.9	0 (0)
Vegetables *n* = 510	315.9	3.3	3.3	3.6	0.7	0.03	392.6	5 (1.0)
Legumes *n* = 100	342.7	2.1	2.1	0.8	0.2	0.03	290.3	0 (0)
Mixed dishes *n* = 299	813.0	3.3	3.2	9.3	4.0	0.17	429.2	10 (3.3)
Protein *n* = 602	787.4	1.4	1.4	9.9	3.5	0.13	826.0	18 (3.0)
Snack foods *n* = 699	2059.4	6.8	6.1	27.9	7.7	0.06	476.8	58 (8.3)
Soups & sauces *n* = 610	676.1	9.7	9.6	11.2	2.2	0.07	746.3	35 (5.9)
Food total *n* = 5290	1072.8	13.4	11.6	12.4	5.1	0.12	411.2	347 (6.6)
Dairy drinks *n* = 306	255.1	6.0	4.9	1.8	1.1	0.07	43.3	58 (19.0)
Other beverages *n* = 478	116.7	5.8	4.0	0.1	0.08	0.004	13.4	213 (44.6)
Sodas *n* = 288	125.1	6.9	6.9	0.04	0.02	0.01	18.6	160 (55.6)
100% fruit juice *n* = 385	190.0	10.4	6.0	0.05	0.02	0.0	9.5	1 (0.3)
Beverage total *n* = 1457	160.7	7.2	5.1	0.45	0.3	0.01	19.7	432 (29.6)
Food & beverage total *n* = 6747	875.7	12.1	10.7	9.8	4.1	0.09	326.6	779 (11.5)

**Table 3 nutrients-13-02584-t003:** Number of packaged SA products that would be regulated by the CWO 2019 NPM criteria (overall, for sugar, sodium, saturated fat and energy).

	Number Regulated (Overall)	Number Regulated for Sugar	Number Regulated for Sodium	Number Regulated for Saturated Fat	Number Regulated for Energy	Number Regulated for Only Energy
Breakfast cereals *n* = 110	97	74	16	31	94	11
Cereals & cereal products *n* = 254	106	1	71	34	47	10
Confectionary & dessert *n* = 1119	1057	997	83	600	912	12
Dairy *n* = 791	515	262	246	56	74	3
Fruits & vegetables *n* = 706	293	129	163	13	29	8
Legumes *n* = 100	28	0	28	0	3	0
Mixed dishes *n* = 299	211	14	177	113	36	3
Protein *n* = 602	412	6	390	88	88	3
Snack foods *n* = 699	564	95	388	394	552	35
Soups & sauces *n* = 610	480	206	416	106	244	12
Food total *n* = 5290	3763	1784	1978	1435	2079	97
Dairy drinks *n* = 306	139	135	4	0	39	1
Other beverages *n* = 478	246	243	1	0	3	2
Sodas *n* = 288	190	190	0	0	2	0
100% juice *n* = 385	6	3	3	0	0	0
Beverage total *n* = 1457	581	571	8	0	44	3
Food & beverage total *n* = 6747	4344	2355	1986	1435	2123	100

**Table 4 nutrients-13-02584-t004:** Pros and cons of different base approaches.

	Pros	Cons
Per 100 g/100 mL	Simple to conceptualize and easy to compare foods Used on nutrition information panels on SA packaged foods	Certain foods are eaten in very small quantities (e.g., oil) while others are consumed in large quantities (e.g., beverages)
Per 100 kJ/% total energy	Allows for food consumed in smaller quantities to be put into context	Difficult to make sense of individual food items that do not represent total energy intake for the day
Per serving	Recognizes that portion sizes of different food types vary significantly, and if eaten in large quantities, will contribute more to nutritional intake than smaller amounts	Serving sizes are determined by the food producer and as a result vary significantly, even within a food category Easy to manipulate serving sizes to appear ‘healthier’, but these are not representative of the amount usually consumed

**Table 5 nutrients-13-02584-t005:** Final proposed cut-points for an NPM suitable to be used in restrictive food policies in SA.

Solid Food (g) Cut-Points		Liquids (mL) Cut-Points	
Sodium mg/100 g	400 mg	Sodium mg/100 mL	100 mg
Total sugar g/100 g	10 g	Total sugar g/100 mL	5 g
Saturated fat g/100 g	4 g	Saturated fat g/100 mL	3 g
Non-sugar sweetener	Contains any	Non-sugar sweetener	Contains any

## Data Availability

Data analyzed for this paper form part of a primary project which is currently being written up in other publications. Data used for this paper will therefore be available upon request and granted for replication purposes. Data are available from the UNC Carolina Digital Repository (https://cdr.lib.unc.edu/). For data inquiries, please contact Donna Miles (drmiles@email.unc.edu) and Jessica Ostrowski (jessica.ostrowski@unc.edu).

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
