# Peer review of "A Fit-for-Purpose Nutrient Profiling Model to Underpin Food and Nutrition Policies in South Africa"

_nutrients, 2021, doi:10.3390/nu13082584_

Round 1

Reviewer 1 Report

This is an excellent paper.

Line 131 needs a minor structural amendment.

Line 166 - there's a repeat of Australia and New Zealand - reword the sentence to avoid repetition

Line 415 - section 3.25. While describing how prolific their use is, this paragraph doesn't describe WHY it is worth restricting the use of Non Sugar Sweeteners (NSS) - what is their potential damage to health? Why should we try to reduce their use. e.g. LIne 447 "Given the current evidence regarding NSS intake"....... you should elaborate what you mean by this.

Overall, I think it's an excellent paper and it is addressing a very complex issue. I don't support the concept of 'nutrients to limit' or 'nutrients to encourage' as I think the totality of the diet MUST be taken into account rather than specific foods. However, the issue of profiling has to be addressed at the same time - it's a difficult conundrum and one this paper addresses well!

Reviewer 2 Report

Referee Report on Manuscripts ID: Nutrients-1270510 “A fit-for-purpose nutrient profiling model to underpin food and nutrition policies in South Africa”

The paper is very well-written and properly structured. But I do have some minor comments, which I outline below:

1.Introduction:

a.The contributions of this study are not clear. What are the main contributions of this study to the existing literature? What kind of knowledge gap that this study is going to fill up?

b.The authors may clearly mention the main objective of this study in the introduction section. They did mention the objective of the study (lines 107-108) under the materials and methods section, which I believe is not the right place.

2.Material and methods:

a.As mentioned above, I would recommend moving lines 107-108 under the introduction section.

3.Data

a.Fix lines 131-132

b.How did the authors collect nutritional information on 6747 packaged foods and beverages? Did the authors hire people? Where are the appendices?

c.Lines: 142-144: “To determine the purpose and target population, we reviewed relevant population-level data and the policy context to identify the key nutritional problems faced by the SA population.” What is relevant population-level data?”Please be specific about the data source.

4.Limitations and strengths of the studya.Incorrect section number. I hope you find my comments helpful in revising this research.
